# Healthcare Service Interventions to Improve the Healthcare Outcomes of Hospitalised Patients with Extreme Obesity: Protocol for an Evidence and Gap Map

**DOI:** 10.3390/mps5030048

**Published:** 2022-06-08

**Authors:** Caz Hales, Rebecca Chrystall, Anne M. Haase, Mona Jeffreys

**Affiliations:** 1School of Nursing Midwifery and Health Practice, Faculty of Health, Victoria University of Wellington, Wellington 6021, New Zealand; rebecca.chrystall@vuw.ac.nz; 2School of Health, Faculty of Health, Victoria University of Wellington, Wellington 6041, New Zealand; anne.haase@vuw.ac.nz; 3Health Service Research Centre, Faculty of Health, Victoria University of Wellington, Wellington 6011, New Zealand; mona.jeffreys@vuw.ac.nz

**Keywords:** extreme obesity, evidence gap maps, healthcare outcomes, hospital, interventions, patients

## Abstract

Hospitalised patients with extreme obesity have poorer healthcare outcomes compared to normal weight patients. How hospital services are coordinated and delivered to meet the care needs of patients with extreme obesity is not well understood. The aim of the proposed evidence gap map (EGM) is to identify and assess the available evidence on healthcare interventions to improve healthcare outcomes for hospitalised patients with extreme obesity. This research will use standardised evidence gap map methods to undertake a five-stage process to develop an intervention–outcome framework; identify the current evidence; critically appraise the quality of the evidence, extract, code, and summarise the data in relation to the EGM objectives; and create a visualisation map to present findings. This EGM will provide a means of determining the nature and quality of health service initiatives available, identify the components of the services delivered and the outcome measures used for evaluation, and will identify areas where there is a lack of research that validates the funding of new research studies.

## 1. Background

### 1.1. Introduction

#### The Problem, Condition, or Issue

Internationally, obesity is common and a major public health concern because of its strong causal association with other health conditions [1,2]. Across the Organisation for Economic Cooperation and Development countries, one in five adults is considered obese, with these rates increasing over the last 40 years. Countries with the highest increases in obesity such as the USA, Aotearoa New Zealand, and Saudi Arabia also have the highest growth in extreme obesity, making up as much as 70–80% of obesity growth [2,3]. Obesity rates are not consistent across all social groups and are affected by gender, ethnicity, and socioeconomic position. For example, in 2021, 5.9% of Aotearoa New Zealand adults were extremely obese, with extreme obesity most prevalent in women (8.3%), Pacific Peoples (24.5%), and Māori (13.0%) ethnicities, and the lowest socioeconomic groups (11.9%) [4].

Of particular interest to this evidence gap map (EGM) are patients with extreme, very severe, or morbid obesity [5,6]. Extreme obesity is typically defined as a body mass index (BMI) equal to or greater than 40 kg/m^2^. Although BMI is frequently used as an indirect measure of obesity, the measurement tool has been criticised for its inability to differentiate between fat and lean mass and the fact that it does not incorporate ethnic differences in the ratio of fat to lean mass [1]; some modified versions of BMI have been proposed for Asian, black, and Arab ethnicities [7]. Similarly, the physical size and shape of the person, critical to determining types of physical support needed, is not taken into consideration when using BMI measurements alone [8]. Therefore, other definitions of extreme obesity are utilised in healthcare to inform service delivery: weighing ≥ 150 kgs, or having large physical dimensions that affect mobility and make moving and handling difficult [9]. 

The prevalence of patients with extreme obesity admitted to hospital is largely unknown, although this is likely to be substantially higher than the general population; one study found the prevalence of extreme obesity in the inpatient medical ward was 16% [10]. Hospitalised patients with extreme obesity have poorer healthcare outcomes compared to normal weight patients [11,12,13,14], with a longer length of stay [13], higher likelihood of intensive care admission [11], increased risk of pressure injuries [14] and falls [12], and a greater risk of readmission within 28 days of discharge [11]. Health services for patients with extreme obesity vary geographically; services do not always meet the care needs of this patient population [15,16,17]; care is often fragmented; and specialised equipment is frequently unavailable, missing, or too small [15,16]. These factors result in healthcare inequities and inconsistency in the delivery of high-quality healthcare. How hospital services are coordinated and delivered to meet patient needs related to extreme obesity is not well understood. In particular, the cost to the health system of poorly delivered services is unknown. 

There is an expanse of literature that aims to capture the cost of treatment for hospitalised patients with obesity, demonstrating in general that as BMI increases, healthcare costs increase [18]. One estimate suggests that for those with obesity—classified in BMI class 1 (obese), 2 (severely obese), and 3 (extremely obese)—healthcare spending increases by 22%, 45%, and 50%, respectively, relative to treatment for those in the normal BMI range. However, it is not clear what contributes to these higher costs. Kent et al. [19] report the greatest medical costs to care for these patients include medications, followed by in-patient care, and ambulatory care within hospital settings. However, given the complexity with which obesity relates to comorbid conditions including depression, coronary heart disease, and type II diabetes, care needs differ substantially across this patient group [18,20]. It is difficult to ascertain whether increased spending is due to a greater number of resources required in treatment, or the need for more expensive, specialised equipment, or both. It also is not clear in which contexts these resources are required, nor for whom and how healthcare services can sustainably deliver to the care needs of this expanding patient group. Scoping the available evidence of the direct costs of healthcare provision for hospitalised patients with extreme obesity is therefore important to fill these knowledge gaps. 

Whilst there is a substantial evidence base supporting health policy related to obesity weight loss management and prevention [1,21,22], there is a paucity of research that specifically relates to health service delivery outcomes for hospitalised patients with extreme obesity. To what extent health service delivery challenges impact on poorer health outcomes of this patient population is largely unknown. Furthermore, the most appropriate measure(s) of health service delivery for this patient group must be better defined, with attention given to how healthcare services can address size-sensitive patient outcomes, which are outcomes directly affected by a person’s physical size and shape [12]. The lack of rigorous and comparable data on the prevalence, demographic, clinical, and service characteristics of people with extreme obesity and evidence on service infrastructure and arrangements that work are impeding the understanding of healthcare quality and how they contribute to health inequity. Furthermore, it limits our ability to adequately plan health and service delivery needs for this increasing patient population.

### 1.2. The Intervention

#### Why Is It Important to Develop the EGM

An evidence and gap map (EGM) is a tool used to inform decision-making and research priorities through the identification of existing evidence, as well as gaps in research [23]. The importance of quality in the delivery of healthcare to improve the ‘health for all’ [24,25] has become a key strategic priority for government and health service agencies [26,27,28]. The EGM aligns with the World Health Organization’s Sustainable Development Goals (SDG) to achieve universal health coverage by providing access to quality essential healthcare services [29]. High quality healthcare services are those that are effective, safe, people-centred, and support the likelihood of desired health outcomes for individuals and population groups [29]. Currently, there are no EGMs that identify and assess the available literature on healthcare service interventions to improve healthcare outcomes for hospitalised patients with extreme obesity. This EGM will provide a means of determining the nature and quality of health service initiatives available, identify the components of the services delivered and the outcome measures used for evaluation, and will identify areas where there is a lack of research that validates the funding of new research studies [30].

## 2. Objectives

The objectives of the map are to: Identify available systematic reviews, primary studies, and impact and outcome-based evaluations of healthcare service interventions for hospitalised patients with extreme obesity.Provide database entries of included studies that summarise the intervention, context, study design, and main results.Identify gaps in the evidence where further primary research is needed.Identify gaps in the evidence related to healthcare equity.

## 3. Methodology

### 3.1. Evidence and Gap Maps: Definition and Purpose

Evidence gap maps are decision-making and research prioritisation tools that ensure existing evidence is available and accessible to researchers and policy-makers to support strategic approaches to new research and policy implementation [30]. EGMs collate evidence of existing and ongoing systematic reviews and primary studies in the form of an interactive map. The map provides a visual summary of existing evidence using a policy or conceptual framework of relevant interventions and outcomes specific to the research topic [30]. We will use standardised evidence gap map methods [23,30,31] to undertake a five-stage process: Develop an intervention/outcome frameworkIdentify the current evidenceCritically appraise the quality of the evidenceExtract, code, and summarise the data in relation to the EGM objectivesVisualise and present our findings

We will use EPPI Reviewer Web software [32] to generate the EGM using the framework depicted below (see Figure 1). 

### 3.2. Framework Development, and Scope

Internationally, healthcare quality and performance indicator frameworks vary in breadth and scope, with frequent overlaps in definitions used to describe the process and outcome domains. Typically, frameworks include both monitoring and improving quality and efficiency of the healthcare system [33,34]. Our intervention and outcomes framework has been developed and adapted from internationally recognised standards and frameworks used for optimising health system performance [24] and measuring quality healthcare [29,33,35]. Whilst quality health services encompass promotion, prevention, treatment, rehabilitation, and palliation across the whole healthcare system, this EGM will specifically focus on the delivery of healthcare services in the inpatient setting. Our nine intervention categories are based on healthcare process measures that support the implementation of quality care. Process measures are used to not only assess the patient’s clinical condition, but also focus on elements of an encounter or episode of patient care [36], thereby informing the categorisation of our interventions. Our six outcome categories are underpinned by the Triple Aim quality improvement framework, which aims to: (1) improve the quality, safety, and experience of care; (2) improve health and equity for populations; and (3) provide the best value for public health system resources [24,33,35].

Six key healthcare measures used to operationally support the aims of the Triple Aim quality initiative include safety, effectiveness, efficiency, timely access, patient experience, and health equity [27]. These outcome measures have been incorporated into our intervention–outcome framework. The framework will be presented in two dimensions, whereby the rows list the interventions, and the columns list the outcome domains. 

### 3.3. Conceptual Framework

Figure 1 illustrates our conceptual framework, where the inputs lead to the intended outcomes and overall impact. Health service delivery interventions and initiatives depend on government, local health authorities, and stakeholder funding, policy, and legislation, in addition to robust research. Through investing in healthcare process interventions that focus on components of a patient encounter or care episode, health services have the potential to improve the quality of care. This conceptual framework includes healthcare process interventions that seek to overcome barriers to healthcare quality and health equity for patients with extreme obesity, such as insufficient resources, education, and workforce development. These healthcare process and outcome measures can be used to inform healthcare purchasing, utilisation, and performance improvements [37]. We will revise our conceptual framework as we develop the EGM. 

### 3.4. Eligibility Criteria

We will include studies that assess health service interventions aimed at improving healthcare outcomes for hospitalised patients with extreme obesity.

### 3.5. Dimensions

The EGM framework developed will be used to inform the inclusion and exclusion criteria for the EGM. 

#### 3.5.1. Types of Study Design

The EGM will include systematic reviews and primary studies that report on health service delivery interventions that aim to improve healthcare outcomes for hospitalised patients with extreme obesity. It will include (a) observational studies, (b) randomised controlled trials, and (c) qualitative studies. We will also include studies published in grey literature such as reports, dissertations, and conference abstracts. 

#### 3.5.2. Types of Settings

We will include studies based in the hospital inpatient setting, such as critical care, acute care, theatres, and rehabilitation where the patient is admitted to hospital for inpatient care. Outpatient services based within the hospital will be excluded. Studies of mixed settings will be included if the intervention primarily occurs in the hospital setting with the target population admitted as an inpatient during the main phase of the intervention. 

#### 3.5.3. Status of Studies

We will include all relevant ongoing and completed studies. Protocols of studies not yet completed will be included.

#### 3.5.4. Population

The target population are hospitalised adult patients (≥18 years) with extreme obesity. For this protocol, extreme obesity is defined as: BMI ≥ 40 kg/m^2^; weighing ≥ 150 kgs, or having large physical dimensions that affect mobility and make moving and handling difficult. 

#### 3.5.5. Interventions

The interventions categories have been developed based on an extensive literature search of best practice evidence and expertise in caring for people with extreme obesity. Our interventions are listed and defined in Table 1.

#### 3.5.6. Outcomes

The outcome categories have been developed based on an extensive literature search of quality care outcome measures and practice expertise in the care of people with extreme obesity. Our outcomes are listed and defined in Table 2.

### 3.6. Search Methods and Sources

We will develop and pilot a search strategy with support from the university subject librarian. This search will be undertaken using the following medical and health databases: MEDLINE (via OVID), Emcare (via OVID), Cochrane Database of Systematic Reviews, CENTRAL, CINAHL (via EBSCOhost), PsychINFO (via Proquest), ClinicalTrials.gov, Dimensions, and the Database of Abstracts of Reviews of Effects (DARE). It will also include social science and policy-relevant databases such as the Campbell Library, ASSIA (via Web of Science), SSCI (via Web of Science), and those of grey literature (Proquest Theses and Dissertation Global, Conference Proceedings Citation Index). We aim to identify ongoing and completed primary studies and systematic reviews. We further aim to capture relevant informal evidence including (but not limited to) clinician notes, and patient and case reports. For a complete example of our search strategy, refer to Appendix A.

### 3.7. Screening and Selection of Studies

Titles and abstracts will be screened for eligibility by two reviewers (C.H. and R.C.) for the first 20% of the abstracts, with disagreements resolved by a third reviewer (either M.J. or A.M.H.). We will use single-reviewer screening for the remaining abstracts and have another reviewer screen all excluded titles, as well as those with any conflicts resolved. Titles and abstracts will be reviewed based on intervention, population, setting, and study design, but not outcome. Full texts of included studies will be double reviewed (C.H. and R.C.), with any disagreements resolved through discussion with a third author (either M.J. or A.M.H.). We will screen the reference lists of included studies identified. The EPPI reviewer software will be used to facilitate the screening process throughout. We will not contact authors regarding missing study information.

### 3.8. Data Extraction, Coding and Management

Coding will be carried out by two reviewers (C.H. and R.C.) for the first 20%, with disagreements resolved by a third reviewer (either M.J. or A.M.H.). We will use single-reviewer coding for the remaining studies. Our coding categories used for data extraction are based on the interventions–outcomes framework (see Table 3). Filters will be added to the map to assist with identification of gaps in the evidence, making the data more accessible to different stakeholder groups. Filters to be included in the map are geographical regions, indigenous populations, health equity, health system funding, study designs, and quality of evidence. We have piloted and tested the extraction form on a sample of studies, which was used to draft the map. Consultation of the draft map was undertaken with stakeholders. Additionally, we will collect data on health equity and will examine if studies assessed the effects of the intervention by health inequality characteristics such as ethnicity/culture, gender, and socioeconomic position.

### 3.9. Quality Appraisal

We will assess the quality of systematic reviews only, using the AMSTAR 2 critical appraisal tool for systematic reviews [38]. Systematic reviews will be given an overall rating of high, medium, or low in terms of the confidence with which their findings can be assured. The results of the critically appraised studies will be shown as a colour code on the map. All systematic reviews will be reviewed independently by two reviewers (C.H. and R.C.), with any discrepancies moderated by a third reviewer (either A.M.H. or M.J.). Studies will not be excluded by their quality.

## 4. Analysis and Presentation

### 4.1. Report Structure

The EGM report that will accompany the online interactive map will follow accepted reporting standards for EGM [31]: executive summary, background, methods, results, and conclusion. We will present any changes made between the protocol and the final report. The results section will present data on the number of studies included from the database search and provide an overview of the types of study designs by intervention, outcomes, and filters used. We will also provide information on how health equity has been considered in the studies, provide an overview of the main gaps in the evidence, and identify any limitations of this research. The conclusions will provide implications for researchers, policy-makers, and healthcare providers, and allow us to propose recommendations for future research priorities.

Tables and figures we will include:Figure: Conceptual framework.Figure: PRISMA flowchart.Table: Number of studies by study design.Table: Number of studies by interventions and outcomes.Table: Number of studies that considered health equity and indigeneity.Other tables and figures will be included based on coded information for selected filters.Appendix: Full search strategy used for each database.

#### 4.1.1. Dependency

We will report studies only once in the EGM. Therefore, studies that include secondary analysis or have published protocols will be reported as one study. We do acknowledge that some studies already presented in the map will be included in one or more systematic reviews.

#### 4.1.2. The Evidence Gap Map

The results of the evidence gap map will be presented as a matrix, with the interventions presented in rows, and the outcomes presented in the columns. We will use bubbles of differing sizes to indicate the volume of studies included. Different types of studies will be presented in different colours. Filters will be included as described above. The online interactive map will be hosted on the Open Access Institutional Repository by Victoria University of Wellington, Aotearoa New Zealand.

## 5. Stakeholder Engagement

Stakeholder consultation is an important part of the methodological process, with the specific purpose of determining the scope of the map, developing the framework, and interpreting the findings [31]. We have created a research network group comprised of healthcare professionals, consumers, and researchers with expertise in obesity, mobilisation, patient injury, health service delivery, health and safety, support work, patient care, healthcare equity, and cultural support. We held an in-person research network meeting in October of 2021 to seek advice and feedback on the scope of the map and the development of our EGM framework. A simultaneous consultation process occurred to seek specific feedback on the alignment of the intervention–outcome framework, with healthcare outcome measures used to evaluate quality healthcare from experts working in health, quality, and safety. These were undertaken via virtual and electronic communication (email consultation, Zoom/Skype meetings). A second network meeting will be held in July of 2022 to review and provide feedback on the mapping of the evidence within the EGM and confirm the interpretation of the findings. Ad hoc consultations will occur with subject experts as and when needed throughout the research process. The research team (C.H., R.C., M.J., and A.M.H.) meet fortnightly to discuss the direction and development of the EGM.

The advisory group members for the EGM are:Dr. Junior Ulu, Director Pacific Peoples Health 2 DHB, Capital & Coast and Hutt Valley District Health Boards.Dr. Aliitasi Su’a Tavila, Senior Lecturer in Pasifika Health, School of Health, Victoria University of Wellington, New Zealand.Levi Vaoga, Healthcare consumer, New Zealand.Tuppy Parker, Whānau Care Services, Capital and Coast District Health Board, New Zealand.Miriam Coffey, Mental Health and Additions Service, Hutt Valley District Health Board, New Zealand.Te Manu Tūtaki, Mental Health and Additions Service, Hutt Valley District Health Board, New Zealand.Catherine Gerrard, Programme Manager, System Improvement, Ministry of Health, New Zealand.Eleanor Barrett, Occupational Therapist, Moving & Handling Advisor, Capital and Coast District Health Board, New Zealand.Dr. Maureen Coombs, Adjunct Professor, School of Nursing Midwifery and Health Practice, Victoria University of Wellington, New Zealand.Ina Farrelly, Podiatrist, Registered Nurse, Health Consultant, Board of Trustee-Tissue Viability Society, United Kingdom.

## Figures and Tables

**Figure 1 mps-05-00048-f001:**
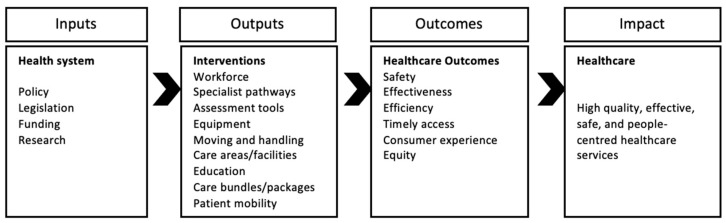
Conceptual framework of interventions and outcomes for quality healthcare.

**Table 1 mps-05-00048-t001:** Intervention categories.

Intervention	Definition
Specialist workforce	Designated health professional roles/positions employed specifically to manage/care for hospitalised patients with extreme obesity. Examples: MeSH Term—“Nurse Specialist” = ‘Nursing professionals whose practice is limited to a particular area or discipline of medicine’ (MEDLINE, Ovid). “Patient Care Team”—‘Care of patients by a multidisciplinary team usually organised under the leadership of a physician’
Special care pathways	Processes that guide the care of hospitalised patients with extreme obesity. Examples: MeSH Term—“Patient Care Planning” = ‘a written medical and nursing care program designed for a particular patient’ (MEDLINE, Ovid)
Assessment tools	Assessment tools used to assess patient risk and guide care during hospital admission. Examples: MeSH Term—“Needs Assessment” = systematic identification of a population’s needs or the assessment of individuals to determine the proper level of services needed (MEDLINE, Ovid)
Equipment	Specialist equipment used/needed in the care of hospitalised patients with extreme obesity. Examples: MeSH Term = “Equipment and Supplies” = Expendable and nonexpendable equipment, supplies, apparatus, and instruments that are used in diagnostic, surgical, therapeutic, scientific, and experimental procedures.’ (MEDLINE, Ovid).
Moving and handling	Processes used during care for the movement and transfer of patients with extreme obesity. The lifting, transferring, repositioning, or mobilising of part or all of a patient’s body. (Focus of the research is on processes to support staff to enable patient mobilisation.) Examples: MeSH Term—“Moving and Lifting Patients” = ‘Moving or repositioning patients within their beds, from bed to bed, bed to chair, or otherwise from one posture or surface to another.’
Specialist care areas	Specific areas/wards where the physical environment has been designed for caring for patients with extreme obesity. Examples: Ward OR Bedspace OR Facilities (Search as keywords in MEDLINE, Ovid)
Education	Bariatric specific education/training designed to improve care delivery for patients with extreme obesity by healthcare professionals. Examples: MeSH Term “Inservice Training”—on the job training programs for personnel carried out within an institution or agency’ (MEDLINE, Ovid)
Care bundles/packages	Groupings of care interventions/practices that, when implemented collectively, improve the reliability of their delivery, and improve patient and healthcare outcomes. Examples: MeSH Term “Patient Care Bundles” = ‘small sets of evidence-based interventions for a defined patient population and care setting’ (MEDLINE, Ovid).
Patient mobility	Patient mobility during their hospital admission. (Focus of the research is on improving the biomechanics of patient mobility.) Examples: MeSH Term “Mobility Limitation”—‘difficulty walking from place to place’ (MEDLINE, Ovid). OR ‘Immobility’, ‘patient mobility’, ‘transfer to sitting’, or ‘transfer to standing’ searched as keywords.

**Table 2 mps-05-00048-t002:** Outcome categories.

Outcome Measures	Subcategory	Definition
Safety	Patient injury	Any hospital-acquired patient injury, adverse event, and care practice that caused harm to the patient.
	Patient falls	Any fall occurring during hospitalisation/hospital stay.
	Staff injury	Any injury or condition experienced during the delivery of care to patients with extreme obesity.
Effectiveness		Providing services based on scientific knowledge to all those who could benefit and refraining from those who would likely not benefit. Outcome data represents either the size of the problem or the reduction in harm. Patient deterioration Hospital-associated infection Medication safety Pressure injuries Falls
Efficiency	Costs	Direct costs of healthcare provision for hospitalised patients with extreme obesity. Costs accrued during an inpatient care episode.
	Resource utilisation	Maximizing the benefit of available resources and avoiding waste. Bed days (intensive care unit length of stay, hospital length of stay) Harm avoidance practices
Timely access		Timeliness of healthcare system’s capacity to provide care quickly after a need is recognised. Reducing waiting times and harmful delays. Time to receive care interventions Waiting times to access services
Patient experience	Satisfaction/experience	The overall patient assessment of the quality of care received during their hospital stay using measures like: being treated with kindness and respect, being listened to, and being involved in decisions.
	Person-centred	Providing care that responds to individual preferences, needs, and values.
	Discrimination	The unjust or prejudicial treatment of the patient with extreme obesity because of their physical health condition or appearance.
Health equity	Quality of care	Providing care that does not vary in quality on account of the patient having extreme obesity. Comparisons between patients who are non-obese and obese for the same intervention
	Access to care	Access to appropriate care based on identified needs of the patient with extreme obesity. Comparisons between patients who are non-obese and obese for the same intervention
	Quality improvement	Improvement efforts aim to improve outcomes for patients with extreme obesity.

**Table 3 mps-05-00048-t003:** Coding categories.

Category		Subcategory
1. Study design		Systematic review RCT Observational studies Qualitative studies
2. Publication status		Complete Ongoing (e.g., protocol)
3. Quality of evidence		High Moderate Low Protocol
4. Region		Europe Oceania North America South America Asia Middle East Africa
5. Setting (Health system funding)		Public Private Both
6. Population		Indigenous-focused Not indigenous-focused
7. Intervention		Specialist workforce Special care pathways Assessment tools Equipment Moving and handling Specialist care areas Education (health professionals) Care bundles Patient mobility
8. Outcomes	1. Safety	Patient injury Patient falls Staff injury
	2. Effectiveness	1. Patient deterioration 2. Hospital associated infections 3. Medication safety 4. Pressure injuries 5. Falls
	3. Efficiency	Cost Resource utilisation
	4. Timely access	1. Time to receive care 2. Waiting times to access services
	5. Patient experience	Satisfaction Person-centred Discrimination
	6. Health equity	Quality of care Access to care Quality improvement
9. Inequities	Is there an assessment of effects by sex/gender?	1. Yes 2. No 3. Planned, but not reported
	2. Is there an assessment of effects by ethnicity/race?	1. Yes 2. No 3. Planned, but not reported
	3. Is there an assessment of effects by socioeconomic position?	1. Yes 2. No 3. Planned, but not reported

## Data Availability

Results data will be hosted on a publicly accessible database.

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
