# Peer review of "Healthcare Service Interventions to Improve the Healthcare Outcomes of Hospitalised Patients with Extreme Obesity: Protocol for an Evidence and Gap Map"

_mps, 2022, doi:10.3390/mps5030048_

Round 1

Reviewer 1 Report

The protocol Healthcare Service Interventions to Improve the Healthcare Outcomes of Hospitalised Patients with Extreme Obesity: Protocol for an Evidence and Gap Map, aims to Identify available systematic reviews, primary studies, and impact and outcome based evaluations of healthcare service interventions for hospitalised patients with extreme obesity.

A very last advice is to recommend the authors to use the term “people with obesity” or “population with obesity” as advised by the latest Guidelines to manage obesity instead of the term “obese” or overweight that may not currently sound politically correct. It is important to make those modifications in the article.

I consider the methodological proposal interesting, but in the last section (5 Participation of the stakeholders), an unclear closure is left of what was achieved in the face-to-face meetings, it is suggested to discuss a little more and what will happen in the meeting in May 2022; the closing of this section is only described about the group formed and not more about what will be done to achieve its objectives, so it is recommended to expand this section.

If these critiques are addressed, then this paper will be ready to be published.  

Reviewer 2 Report

The manuscript is a protocol about the methods for a systematic review of the literature looking for the identification of gaps that would implemented the treatments of patients with extreme obesity. I think the paper is well written and the goals are clear. The authors have included all the methodological aspects that a systematic review requires. I have only few comments:

- please introduce all the acronyms used (for example OECD, line 33)

- I think the psychological/psychiatric factors are quite underestimated in the overall methods. Please evaluate the inclusion of elements like bodyimage concerns or psychiatric conditions. Indeed, intervention for extremely obese patients has been linked to body image changes. 

- it is not clear to me the role of the advisory panel member at the end of the paper. Should they be moved to the acknowledge paragraph?
